# Surface atmospheric forcing as the driver of long-term pathways and timescales of ocean ventilation

Alice Marzocchi[1], A. J. George Nurser[1], Louis Clément[1], and Elaine L. McDonagh[1,2]

[1]National Oceanography Centre, Southampton, United Kingdom
[2]NORCE, Norwegian Research Centre, Bjerknes Centre for Climate Research, Bergen, Norway

**Correspondence:** Alice Marzocchi (alice.marzocchi@noc.ac.uk)

**Abstract.** The ocean takes up 93% of the excess heat in the climate system and approximately a quarter of the anthropogenic carbon via air-sea fluxes. Ocean ventilation and subduction are key processes that regulate the transport of water (and associated properties) from the surface mixed layer, which is in contact with the atmosphere, to the ocean's interior which is isolated from the atmosphere for a timescale set by the large-scale circulation. Using numerical simulations with an ocean-sea-ice model using the NEMO framework, we assess where the ocean subducts water and thus takes up properties from the atmosphere and how ocean currents transport and redistribute them over time and how, where and when they are ventilated. Here, the strength and patterns of the net uptake of water and associated properties are analysed by including simulated sea water vintage dyes that are passive tracers released annually into the ocean surface layers between 1958 and 2017. The dyes' distribution is shown to capture years of strong and weak convection at deep and mode water formation sites in both hemispheres, especially when compared to observations in the North Atlantic subpolar gyre. Using this approach, relevant to any passive tracer in the ocean, we can evaluate the regional and depth distribution of the tracers, and determine their variability on interannual to multidecadal timescales. We highlight the key role of variations in subduction rate driven by changes in surface atmospheric forcing in setting the different sizes of the long-term inventory of the dyes released in different years and the evolution of their distribution. This suggests forecasting potential for determining how the distribution of passive tracers will evolve, from having prior knowledge of mixed-layer properties, with implications for the uptake and storage of anthropogenic heat and carbon in the ocean.

## 1 Introduction

The ocean absorbs more than 90% of anthropogenic warming and is the largest, mobile carbon reservoir in the climate system that is accessible on millennial timescales. Ocean ventilation is the process by which air-sea fluxes of properties such as heat and carbon penetrate into the enormous reservoir that is the ocean interior. Waters recently exposed to the atmosphere, that reside in the ocean's surface mixed layer, pass into the ocean's interior; this is balanced by entrainment of waters from the ocean interior into the ocean mixed-layer (Stommel, 1979). The values of mixed layer properties such as temperature and carbon content depend on the rate of this exchange (Abraham et al., 2013; Banks and Gregory, 2006; Iudicone et al., 2016), as well as the uptake from the atmosphere. Indeed, the atmospheric uptake of heat and carbon itself also depends on ocean

ventilation through this impact of exchange on mixed-layer temperature and carbon. Consequently, ocean ventilation plays a key role in modulating climate variability on interannual to decadal (and even centennial) timescales.

Ventilation involves exchange between the surface mixed-layer and the ocean interior (Luyten et al., 1983). This exchange is effected in reality on all time and space scales, ranging from the global-scale meridional overturning to small scale turbulent exchange. The larger scale exchanges involve the net-annual transfer of fluid from mixed-layer to interior (the net subduction) while the smaller-scale exchanges are associated with geostrophic eddies and small-scale turbulence. In ocean models these smaller-scale exchanges are parameterised as diffusive fluxes. Both large-scale "advective" and small-scale "diffusive" exchanges are key drivers of the uptake of anthropogenic heat and carbon by the ocean (Williams and Meijers, 2019) and their observed signals have been estimated by the distribution of transient tracers (e.g. chlorofluorocarbons - CFCs - and tritium).

It is not only the rate of exchange between ocean surface mixed layer and interior that is important, but also the length of time that subducted waters remain in the interior before coming up again and being re-entrained/obducted into the surface mixed layer. For example, as a consequence of anthropogenic warming the subducted waters warm with time as the mixed-layer warms, so where these warmer waters re-entrain into the surface mixed-layer rapidly within a few years of their subduction the surface mixed-layer will warm further. Conversely, waters that spend longer in the interior before they obduct will have subducted earlier when they are still relatively cool, and so these warm the mixed-layer less when they re-entrain.

The fidelity of future climate projections relies on an accurate representation of ocean ventilation and on the ability of the next generation of numerical models to predict transient and regional climate change (Katavouta et al., 2019). Climate simulations show wide variations in their representation of processes critical to ocean ventilation and its variability, such as mixed layer depths in the subpolar gyres, and the strength, depth and variability of the Atlantic Meridional Overturning Circulation (AMOC) (Heuzé, 2017, 2021). Consequently, variable representation of the ventilation process is a major source of intermodel spread in the projection of carbon and heat uptake (Boé et al., 2009; Frölicher et al., 2015) by the ocean, and consequently regional sea level rise (Kuhlbrodt and Gregory, 2012; Church et al., 1991).

The high-latitude oceans (e.g. subpolar North Atlantic and Southern Ocean), where the densest waters are formed, play a prominent role in global ventilation, since up to two-thirds of the volume of the ocean interior and more than three-quarters of the deep ocean are thought to be ventilated at these locations (Khatiwala et al., 2012). Many of these deep waters are ventilated on long timescales and a substantial fraction of the deep ocean is ventilated in the high-latitude North Atlantic, feeding the lower limb of the AMOC (Lozier, 2010, 2012). Consequently, this region plays a prominent role in transient climate change, both through the uptake and redistribution of heat, and the long-term sequestration of anthropogenic carbon (Zanna et al., 2019; Khatiwala et al., 2009).

Differently from the more established view, recent studies using new observations in the subpolar North Atlantic, indicate the prevailing importance of the deep waters formed in the Irminger and Iceland basins over the Labrador Sea as largely responsible for the observed variability in the overturning (Lozier et al., 2019; Li et al., 2019; Zou et al., 2020; Petit et al., 2020). The link between the overturning and the buoyancy forcing in the subpolar gyre points to the important role played by surface forcing (i.e. air-sea fluxes) in establishing the state of the AMOC and suggest the need to investigate the sensitivity of the AMOC to the observed interannual variability observed in these regions (Petit et al., 2020).

Processes occurring at lower latitudes and in upwelling layers also affect the renewal of water masses in the ocean's interior, indicating a decoupling of ventilation from the overturning circulation. Subduction into the subtropical thermoclines is a major driver of ventilation. Additionally, diapycnal and isopycnal diffusion play an important role in ventilation both at low and high latitudes e.g. in the upwelling regions of the Southern Ocean (Naveira Garabato et al., 2017). This means that dense water formation at high latitudes (and the overturning circulation) may depend on different dynamics and occur at different locations and on different timescales than ocean ventilation, which largely depends on the connection of tracers to the surface ocean (MacGilchrist et al., 2020).

Age tracers that track the length of time since waters have been ventilated have previously been used in numerical models to understand ventilation timescales (England, 1995; Haine and Richards, 1995; Hall et al., 2002) and Lagrangian particle tracking experiments (Van Sebille et al., 2018) have been performed to investigate high-latitude ventilation with a focus on the North Atlantic subpolar gyre (e.g. MacGilchrist et al., 2017, 2020, 2021). However, both of these approaches have limitations. Since any "water parcel" is actually made up of a mixture of waters from different sources, in general it has an age distribution rather than a specific single age, while Lagrangian particle tracking cannot fully account for the effects of mixing and diffusive processes. In addition, observational analysis is often based on the interpretation of the distribution of passive transient tracers, such as CFCs, but while the interpretation accommodates advective and diffusive processes, it does generally assume a steady state background ocean circulation.

We carried out numerical simulations where we resolve the interannual variability in both forcing and circulation, and discern the impact that this variability has on the inventory and distribution of a passive tracer. To achieve this, we analyse changes in ocean ventilation using simulated interannually-varying dye tracers, which represent distinct annual sea water vintages. These allow us to explore the links between subduction, ocean circulation and surface forcing on a range of different timescales and globally, by following the pathways of the passive tracers. With our approach, we aim to separate the roles played by deep water formation, ventilation, overturning and how these are driven by surface forcing on interannual to interdecadal timescales.

Our simulations can be used to inform the interpretation of anthropogenically-sourced transient tracers such as CFCs (e.g. Haine and Richards, 1995; England, 1995; Hall et al., 2002; Fine et al., 2017) and other derived quantities such as anthropogenic carbon (Khatiwala et al., 2009, 2012, 2013) and excess heat (Banks and Gregory, 2006; Zanna et al., 2019; Zika et al., 2020). This experimental design (see section 2.3) is complementary to Green's function approaches (Haine and Hall, 2002; Primeau, 2005), where there is an assumption about a steady-state circulation and times and locations of ocean ventilation are expressed as probability distributions. Our simulated dye tracers effectively represent an "explicit" Green's function, where the ocean circulation is time-varying and which allows us to estimate the evolution of the pathways and timescales of ocean ventilation. However, our simulations are computationally expensive to run, limiting the horizontal resolution that we can use, since our aim is to generate a set of interannually-varying tracers for 60 years of available atmospheric forcing (i.e. 60 tracers for each year of simulation).

The model, its spin-up, and the experimental design for the injection of the dye tracers are described in section 2. Changes in the dye distribution are shown globally and as a function of latitude and depth in section 3, highlighting the role of interannual variability in the evolution of the different dyes and differences/similarities between the two hemispheres. The dominant role

of surface forcing in setting these patterns and its implications for the interpretation of observational data, as well as potential
drivers of longer-term changes and larger-scale signals, are discussed in sections 4 and 5.

## 2 Methods

### 2.1 Ocean-ice model

Numerical simulations are run using a global configuration of the Nucleus for European Modelling of the Ocean model s
(NEMO Madec, 2014) ocean model, which is coupled to the Los Alamos sea-ice models (CICE Rae et al., 2015). We use
ORCA1, which has a horizontal resolution of $1°$ (approximately 73 km) and 75 vertical levels, where the thickness increases
vertically, from 1 m at the surface to 200 m at depth. We chose not to use a higher resolution version because of the heavy
computational requirements of the simulations. More complete descriptions of ORCA1 and a comparison of its performance
with configurations with different horizontal resolutions, can be found in Storkey et al. (2018) and Kuhlbrodt et al. (2018).
ORCA1 also makes up the physical ocean component of the coupled UK Earth system model (UKESM1), contributing to the
sixth Coupled Model Intercomparison Project (CMIP6), as described in Yool et al. (2020).

This model configuration involves a vertical mixing scheme based on the turbulent kinetic energy model of Gaspar et al.
(1990), together with Redi isoneutral mixing (Redi, 1982) with diffusivity of 1000 $m^2$ $s^{-1}$ and Gent and McWilliams (1990)
eddy parameterization with a coefficient dependent on the local Rossby radius and Eady growth rate (Held and Larichev,
1996). Temperature, salinity and tracers are advected by a second-order, two-step monotonic flux corrected scheme (FCT)
Zalesak (1979), which is the standard scheme (Lévy et al., 2001) used in current low-resolution NEMO models (UKESM1;
e.g. Yool et al., 2020). The numerical mixing associated with this scheme may lead to an overestimate of Antarctic Bottom
Water (AABW) production (Hofmann and Morales Maqueda, 2006) compared with more accurate but expensive advection
schemes, such as the 2nd-order-moment scheme (Prather, 1986), which is, however, not implemented in NEMO.

Our ocean-ice simulation is forced with the JRA55-do surface atmospheric dataset (Tsujino et al., 2018) that is based on the
Japanese 55-year Reanalysis (JRA-55 Kobayashi et al., 2015) and it is spun up following the Ocean Model Intercomparison
Project (OMIP) protocol (Griffies et al., 2016), which recommends running five cycles of the atmospheric forcing, starting
from a motionless ocean state with climatological temperature and salinity distribution. Each cycle lasts 60 years (the length
of the available atmospheric forcing for the version used of JRA-55, from 1958 to 2017), and our diagnosis focuses on the last
(fifth cycle), with the previous four cycles (240 years) used for spin-up. This OMIP protocol allows for meaningful comparison
between different models (e.g. Danabasoglu et al., 2014), although there may still be drifts in many ocean properties (Griffies
et al., 2016). All physical variables and those for the dye tracers are output and stored as monthly means.

Our model configuration gives an average AMOC (maximum in the overturning streamfunction at 1000 m at $26°N$) of $\sim12$
Sv (where 1 Sv = $10^6 m^3 s^{-1}$) in the last 60 years (1958-2017) of the spin-up, which is lower than the latest observation-based
estimates from the RAPID array at $26.5°N$ (17-20 Sv Moat et al., 2020), but is not uncommon given the relatively coarse
horizontal resolution of the model (e.g. Storkey et al., 2018; Hirschi et al., 2020). The average transport through Drake Passage
is also weaker ($\sim115$ Sv) than the observation-based estimates of $173 \pm 11$ Sv (Donohue et al., 2016). Both the average AMOC

and Drake Passage transport are weaker in our ocean-only simulation than in the coupled UKESM1 version of the model (Yool et al., 2020).

## 2.2 Mixed layer depth and transient tracers

The representation of the mixed layer in the model is of particular interest for our analysis, given its key role in driving ocean ventilation (Pedlosky and Robbins, 1991). Therefore, we compare this simulated quantity from the NEMO output with the objectively-analysed EN4 dataset (Good et al., 2013) and an optimally-interpolated Argo dataset (King et al., 2021). For all datasets and for the time period 2004-2017, the maximum mixed layer depth (Figure 1) is calculated at each gridpoint, using a variable density threshold associated with a $0.2°C$ decrease in temperature, and representing the depth of the base of the winter
mixed layer, as defined in (Clément et al., 2020).

The model reproduces the deep winter mixed layers both in the North Atlantic subpolar gyre and in the sub-Antarctic mode water formation regions, as well as along the Antarctic coast (Figure 1a) which is not captured in the EN4 and Argo datasets (Figures 1b,c,d) due to limited observational capability under ice. However, it is known that the NEMO model is characterised by excessively strong convection in the Labrador and Irminger Seas, down to depths that are much higher than
140 what is measured in observations (Figure 1c), even at higher horizontal resolutions (MacGilchrist et al., 2020; Marzocchi et al., 2015; Rattan et al., 2010) and overproduction of Labrador Sea Water is also found in other $1°$ ocean-ice models when compared to observations (Li et al., 2019). This results in some overestimation in the mixed layer depth in the North Atlantic subpolar gyre, but globally the model shows good agreement with observations; deep mixed layers can also be observed along the pathway of the Gulf Stream and Kuroshio Current, both in the model and in the observations, although in the Atlantic
between ∼40-60°N the simulated mixed layer is largely shallower than in the observations (Figure 1c).

Anthropogenic transient tracers such as CFCs are often used for model validation and to evaluate ocean ventilation (Orr et al., 2017; Fine et al., 2017; Dutay et al., 2002). These remain inert when absorbed by the ocean from the atmosphere and due to their passive nature they can be thought of as dye tracers. CFC-11, CFC-12 and SF-6 tracers are included in our simulations and here we show a comparison of their concentration along a mid-Atlantic section (Figure 2) that will be used in the rest of the
150 analysis. The tracers are implemented following the OMIP protocol (Orr et al., 2017) and have also been used for the evaluation of the coupled UKESM1 simulation (Yool et al., 2021). The CFC-11 distribution in our simulations compares favourably to observations from the Global Ocean Data Analysis Project (GLODAPv2.2020; Olsen et al., 2020) along the WOCE/GO-SHIP A16 section (from occupations in 2003, 2005, 2013, 2014; cruises 342, 343, 1041, 1042) in the Atlantic Ocean (Figure 2). The A16 section crosses the Atlantic from south of Iceland to the Southern Ocean and covers the eastern part of the basin in the
155 Northern Hemisphere and the western side in the Southern Hemisphere, while the section in the model is taken in the central part of the basin at 25°W. Both in the model and in the observations, the highest concentrations can be found at high latitudes in both hemispheres, where CFC-11 has spread into the ocean's interior, especially in the subpolar North Atlantic (below 2000 m). The model also appears to be capturing the penetration of the tracer at depth in the Southern Hemisphere (below 4000 m), even though here the concentrations are much lower in the model than in the observations. The tracer also penetrates down to
160 ∼ 2000 m around the Equator in the observations (Figure 2b), but this is not captured in the simulation.

It is also worth noting that there are large differences in the simulation of the magnitude and variability of the (A)MOC in numerical models, both at lower and higher resolutions (e.g. Hirschi et al., 2020; Li et al., 2019). There are also substantial discrepancies in simulating the mixed layer depth, especially in high latitude regions, and in the representation of water masses, as clearly shown for Labrador Sea Water (Li et al., 2019). Climate models are also known to struggle with the representation of dense water formation in the Southern Ocean, especially at lower resolutions (Heuzé et al., 2013), which will have an impact on the rates and dynamics of ventilation.

## 2.3 Dye tracer injection

In this study we aim to resolve the pathways and the inventories of ventilated waters after they leave the mixed layer. This is achieved by using a set of annually-distinct dye tracers that are introduced in each year of the last (fifth) cycle through the surface forcing, meaning that the effective model spin-up is 240 years and the simulation with passive tracers is 60 years, which is the length of the available atmospheric forcing for the version used of JRA-55 (from 1958 to 2017).

In these simulations, the dye tracers are released uniformly across the global ocean, with a 6-month hemispheric offset, since tracer injection starts during the summer in each hemisphere (i.e. starts in January and ends in December in the Southern Hemisphere and starts in July and ends in June in the Northern Hemisphere). A transition zone is introduced between 20°N and 20°S, where the start and end of the release year increase day by day linearly moving northwards; i.e. the release year at a given latitude begins on the year day-number (days from 1 January) that increases linearly between 20°S and 20°N from zero to 181 (the number of days between 1 January and 1 July), and similarly for the end of the release year.

Each dye tracer is injected throughout its release year (its 'vintage') by relaxing the tracer's concentration to a value of 1 throughout the top seven vertical levels (down to ∼10m). The relaxation time is $T_r = \frac{T_{r0}}{1-a_i}$, where $T_{r0}$=7200 s (the leapfrog timestep) is the smallest relaxation time that still remains stable. To represent the ability of ice cover to insulate the ocean from air-sea gas exchange, the relaxation time is increased by the reciprocal of the water fraction 1-$a_i$ with $a_i$ the ice fraction. With the depth interval of 10 m, this implies a piston velocity of 1.39 x $10^{-3}$ m s$^{-1}$, (120 m day$^{-1}$) without ice (much faster than a typical gas exchange piston velocity of 2.4 m day$^{-1}$), ranging down to a velocity of 2.4 m day$^{-1}$ in regions of maximum ice-cover with $a_i$=0.02. After injection into the interior, each tracer continues to propagate through the ocean, driven by advective and diffusive components of the simulated circulation. After its release year, each tracer is relaxed back to zero in the upper 10 m, resulting in a systematic loss of the globally integrated tracer inventory over the simulation. This is presented schematically in Figure 3, where dye N is only injected in release (vintage) year N, while dye N+1 only starts being injected in release year N+1, and so on for the following years (from N+2 to N+59).

For the analysis presented in section 3, we use a subset of dyes, injected in 1958 to 1993, so that we can follow their evolution from the year of injection and for 25 years of simulation for each one of the 36 vintages, allowing us to investigate the role of interannual variability and pathways from annual to multidecadal timescales. Note that the comparison of these simulated dye tracers to observed CFCs distributions (see Figure 2) is not straightforward. While the CFCs source behaves like a step function, our vintage tracers are best represented by a top-hat/delta function, so it is the sum of the concentration of all the different vintage tracers that could be compared with CFCs.

## 2.4 Dye diagnostics

The dye concentration, c, at any point (or in the model, in a gridbox) represents the fraction of water at that point that was exposed to the surface in the dye's release winter (vintage). The natural unit for the dye concentration is, therefore, dimensionless (a fraction, so $0 < c < 1$). Then the total globally-integrated inventory of dye $\mathbb{C}_{global} = \iiint c \, dV$ represents the total volume of water exposed to the surface in the release year. Similarly, the 'ventilation thickness' $C(\phi, \lambda) = \int c \, dz$ (where $\phi$ and $\lambda$ are latitude and longitude) represents the volume of such ventilated waters per unit area and has dimensions of depth (m), even though it is not a physical depth but the column integral of the tracer. This metric can be thought of as the depth of a notional layer in which the vertically-integrated inventory is concentrated with a dye concentration of c=1. In the first year this represents the mixed-layer depth as c≈1 in the mixed-layer and c≈0 below it. We find it useful to define globally- (and hemispherically-) averaged ventilation thicknesses that relate to the globally- and hemispherically-integrated inventories. In subsequent years, this ventilation thickness progressively decreases as the dye re-enters the mixed layer and is lost (set to zero).

## 3 Results

Our setup allows us to identify when and where water masses were last ventilated and to investigate the role of interannual variability in determining the tracers' distribution, as well as unpicking the role of surface forcing and ocean circulation in setting these pathways and timescales.

### 3.1 Global dye distribution

The dye inventory evolves differently over time in each basin and most of it can be found in the Atlantic on the time scales that we are examining, despite the substantially smaller volume of the Atlantic than the Indo-Pacific (Figure 4). Over 60 years, the percentage of the global inventory held in the Atlantic increases. This is because in the Atlantic the dye penetrates more deeply to depths where it is unable to re-enter the surface mixed layer, while the penetration into the deep Southern and Pacific oceans is less marked (Figure 5a,b). This may not be fully realistic; as previously discussed, models are known to struggle with the representation of dense water formation in the Southern Ocean, especially at the coarse resolution of this simulation, which may explain the low concentrations (and relatively small inventory) of tracer in the deep Southern Ocean (Figures 4 and 5).

Within 60 years (the end of the simulation), the core of the dye in the Atlantic has reached the deep ocean (down to ∼3000 m) between 20 and 50°N, while in the top 1000 m it has spread to ∼40°S (Figure 5a). Looking at the dye distribution as a function of depth clearly indicates how important the Atlantic is for ventilating the deep ocean, especially on these time scales (Figure 5b). The latitudinal distribution of the dye (Figure 5c) illustrates the importance of the subtropical gyres, which on these time scales (60 years after injection) hold most of the dye concentration, especially in the Northern Hemisphere, with the North Atlantic clearly dominating the global zonal average.

In our setup (see Figure 3) we can also analyse the evolution of dyes injected in each year (vintages) separately. This can be effectively displayed using the 'ventilation thickness' metric, as defined in Section 2.4 (Figure 6). For each year, the dye concentration peaks during the winter in each hemisphere (where it is injected) and the ventilation thickness in the year of injection is always higher in the Southern Hemisphere (Figure 6c). In each hemisphere, the interannual differences between each of the dyes largely derives from changes in the background ocean circulation and the effect of surface forcing on the mixed layer depth at the time of injection, as the dyes are independent of each other and identify different sea water vintages for each year. After the first year of dye injection, the interannual variability in the peak in ventilation thickness between the different vintages mostly reflects the variability in mixed layer depth in different years, driven by the surface forcing (Figure 6).

Globally, during the first year when the dye is injected, the highest tracer concentrations can be seen in regions that correspond to the deepest mixed layers (see Figure 1a), such as the subpolar North Atlantic near the locations of deep-water formation from the Labrador and Irminger Seas to the Nordic Seas, as well as along the paths of the Gulf Stream and Kuroshio and in the Mediterranean Sea (Figure 7a). Strong dye uptake can also be seen in the Subantarctic mode water (SAMW) formation regions in the Southern Hemisphere and along the Antarctic coast and in the Weddell Sea, where dense water is formed (Figure 7a). For the rest of the analysis, the main focus will remain on two chosen timescales, of 3 and 25 years, by using a subset of the dye tracers (as introduced in section 2.3). Three years after injection, the vintages represent seawater that has moved below the base of the mixed later and been subducted, while after 25 years they correspond to water that has flowed out of the upper ocean. Three years after injection, the dye is spreading prominently in the North Atlantic subpolar gyre (Figure 7b) and after 25 years it has reached the entire basin, while it is also starting to spread southward along the western boundary (Figure 7c). It is, however, worth remembering that boundary currents are not particularly well-represented at $1°$ horizontal resolution, which will affect the dye distribution (e.g. England, 1995). In the Southern Hemisphere, the uptake of dye can still be clearly identified in the SAMW formation regions after 3 and 25 years (Figures 7b,c). Concentrations in the Arctic are likely too high (Figures 7b,c), possibly as an artifact due to the model's resolution, meaning that the dye accumulates in this region due to restrictions in the flow through bathymetric features that are poorly represented at $1°$. However, these concentrations decrease over time, as the dye excess is slowly being ventilated out of the basin (not shown) which has a relatively small volume and a small contribution to the global inventory (see Figure 4).

### 3.2 Interannual variability in ventilation thickness and ventilation timescales

By using a subset of dyes, from 1958 to 1993, we can follow their evolution from the year of dye injection and for 25 years of simulation for each one of these 36 vintages. Looking at the evolution of the dyes as the anomaly from the mean ventilation thickness (for the 1958-1993 time period) highlights the differences between the independent vintages that are due to interannual variability (Figure 8). The strongest differences from the mean (up to $\sim$10 m in the Southern Hemisphere) develop within the first three years, but these reduce substantially or at least stabilise by three years after injection in both hemispheres. However, after three years of the simulation the differences remain relatively large in the Northern Hemisphere (Figure 8a), while they fall off in the Southern Hemisphere (Figure 8b). After 25 years, the ventilation thickness still varies by up to $\pm$3 m

between vintages in the Northern Hemisphere, while in the Southern Hemisphere differences reduce to within $\sim\pm1.5$ m (Figure 8). Positive anomalies (i.e. ventilation thickness higher than the mean) characterise the later years/vintages (1980s/1990s) in the Northern Hemisphere, while this is opposite in the Southern Hemisphere, where this is the case for earlier years (1960s); similarly, for the negative anomalies (Figure 8).

For all vintages, the absolute ventilation thickness is higher in the Southern Hemisphere after three years than in the Northern Hemisphere (Figure 9). Vintages with a ventilation thickness that is higher than the mean are not necessarily the same in both hemispheres, representing the different response to the surface forcing and the background circulation (e.g. stratification) and local process at play. On this time scale, the vintages represent seawater that has been subducted after injection and then has moved below the base of the mixed later. In the Northern Hemisphere, the vintages from the early 90's have relatively high values (Figure 9a), which correspond to years of observed strong convection in the Labrador Sea (e.g. Yashayaev, 2007; Rhein et al., 2017). Values in the Northern Hemisphere are actually above or very close to the mean for the entire period from the early 80's to the early 90's, while in the Southern Hemisphere ventilation thicknesses higher than the mean can mainly be seen for the earlier period until the late 70's (Figure 9a). Even though the mean hemispheric ventilation thickness values are much closer to one another 25 years after injection, the early 90's vintages in the Northern Hemisphere are now the highest, as well as for the early 80's (Figure 9b), which also stood out at the 3-year timescale. This suggests that the response after 25 years, when the vintages represent water that has now flowed out of the upper ocean, is still strongly affected by the processes shaping the mixed layer at the time of dye injection and for the subsequent 2-3 years.

To further investigate this apparent link between the processes affecting subduction close to the time of dye injection and over the following two decades, we correlate the ventilation thickness for the two timescales of 3 and 25 years (Figure 10). These show high correlation ($R^2 \geq 0.7$) for both hemispheres, which confirms that the conditions close to the time of injection (i.e. mixed layer depth and background circulation) are driving the amount of seawater being subducted, but also that this signal persists over time and that its inventory over 25 years is strongly related to that initial forcing.

Ventilation in the Northern Hemisphere appears to be twice as persistent as in the Southern Hemisphere (see the slope for the correlations in Figures 10a,c), which could be thought of as a rate of erosion of the ventilated water masses. This means that subducted waters are exported (and isolated) away from deep mixed layer regions faster in the Northern than the Southern Hemisphere. The residuals of the correlations (Figures 10a,c) also highlight some of the longer-term variability, particularly in the Northern Hemisphere, where the residuals for the "later" vintages corresponding to the early 90's appear to be rising (Figure 10b).

### 3.3 Longer-term effect of the initial surface forcing on dye distributions

Dyes that were injected in years characterised by stronger convection result in higher ventilation thickness after 3 and 25 years (Figure 9), but this also impacts the evolution of the tracer's vertical distribution over time. This is shown for seven of the vintages (highlighted in Figure 10) analysed so far, covering a range of higher and lower (than the respective hemispheric mean) ventilation thicknesses (Figures 11 and 12).

The vertical distribution of the tracer (Figure 11a) is characterised by two prominent peaks, where the shallower one at around 500m is present in all ocean basins but dominated by the inventory in the Pacific (not shown). The deeper peak below 2000m is driven by the variability in the North Atlantic (not shown) and in fact the highest values are reached in 1992 and 1982. The strongest differences in the tracer's latitudinal distribution between different vintages develop in the Northern Hemisphere subtropical gyres (Figure 11b) and are dominated by the North Atlantic (not shown), which can be expected on these time scales (i.e. 25 years after injection). The 1992 vintage stands out and it corresponds to a period of strong convection in the Labrador Sea, as discussed in section 3.2 and shown in Figure 9, but 1982 and 1972 also have a relatively high peak around 40°N. In the Southern Hemisphere, there are smaller differences between the different vintages, but as expected (see Figure 9a) the highest peak corresponds to the dye injected in 1962 (dominated by the signal in the Pacific, not shown).

Note that Figures 11 a and b and equivalent to Figures 5 b and c, respectively, but here the different dyes are shown 25 years after injection (too allow the comparison between different vintages), while Figure 5 shows the dye distribution at the end of the simulation, 60 years after dye injection.

Differences in the Atlantic are also shown along a north-south section in the centre of the basin (same as Figures 2a and 5a) for two of the vintages 25 years after their injection (Figure 12), one from a strong Northern Hemisphere ventilation year (1992) and a weak one (1987). The tracer reaches deeper and with higher concentrations when they are injected in years characterised by strong convection in the subpolar North Atlantic (Figure 12a) and the distribution is also lower in the top ∼1000 m both in the Northern and Southern hemispheres for the year with weaker convection (Figure 12b).

## 4   Discussion

When considering ocean ventilation and how it regulates the dynamics of tracers and their connection to the surface ocean, it is important to remember that it is driven by processes occurring at lower latitudes, as well as the overturning circulation and dense water formation in the high-latitude oceans (Naveira Garabato et al., 2017; MacGilchrist et al., 2020). In particular, subduction into the subtropical thermoclines drives substantial ventilation. Additionally, diapycnal and isopycnal diffusion play an important role in ventilation both at low and high latitudes (Naveira Garabato et al., 2017). These aspects are included in our simulations, by using passive tracers that reproduce both the advective and diffusive components of the circulation, and this is highlighted by our analysis of interannual to multi-decadal subduction and ventilation.

Despite the known biases for NEMO, such as the too-deep mixing in the Labrador Sea (as discussed in section 2.2) and the fact that not all processes can be simulated accurately at resolutions that are not eddy-resolving, this does not affect our results correlating ventilation thickness on different timescales and highlighting the dominant role of surface forcing in setting the evolution of the tracers' distribution and inventory.

### 4.1   The fingerprint of surface atmospheric forcing

The dye uptake in the first (injection) year largely reflects the state and variability of the mixed layer during the season of active convection, which is driven by the surface forcing in that year (Figures 1a and 7a) and the stratification in that year

("preconditioning"). Three years after injection, the amount of dye that has reached below the base of the mixed layer is also
dependent on other factors, linked to both lateral mixing and the surface forcing in the years immediately following injection
(see also MacGilchrist et al., 2021). This determines how much dye is retained, since a deepening of the mixed layer in the
winter following the injection year (N+1; see Figure 3) will ventilate a larger portion of the vintage from the previous year (N;
see Figure 3) and effectively reduce the dye concentration (since in our setup it will be reset to zero when ventilated). This
contributes to the interannual variability on this timescale.

The correlation between dye retention after 25 years and the background conditions close to the time of injection (just after
the strongest interannual differences have decreased; see Figure 8) highlights the key role of the surface atmospheric forcing
in driving long-term ocean ventilation, and more broadly, in determining the distribution of passive tracers over time (Figure
10). Since the variability in ventilation near the time of dye injection sets the long-term variability for the dye inventory, there
is potential for forecasting how the distribution of a tracer in the ocean will evolve in the future, from a prior knowledge of the
surface air-sea fluxes and mixed-layer properties.

Finally, the strong correlations in ventilation thickness between 3 and 25 years after injection in both the Northern and South-
ern hemispheres (Figure 10; see slope of correlations) imply that, given the strong interannual variability in the initial surface
forcing, it is this variability that will continue to dominate on longer time scales, largely overriding the different processes that
drive how passive tracers are removed or taken up in the two hemispheres. The Northern Hemisphere is characterised by more
persistent anomalies, since the ventilated waters penetrate more deeply where they are better isolated from surface influence
while in the Southern Hemisphere the anomalies are initially stronger (Figure 8), but then dissipate faster than in the Northern
Hemisphere, partly due to the more effective mixing along sloping isopycnals in the Southern Ocean. This means that the tracer
eventually gets mixed back and ventilated even when the initial amount that is subducted is substantially higher than the mean,
while the dye remains in the interior for longer in the subpolar North Atlantic once it has reached a deeper horizon (Figures
11 and 12). In other words, subducted waters are exported (and isolated) away from deep mixed layer regions faster in the
Northern than the Southern Hemisphere.

## 4.2  Longer-term and large-scale signals

The strong interannual variability that characterises ventilation in the Northern Hemisphere on these timescales (Figure 9) is
largely driven by the Irminger, Nordic and Labrador Seas (Figures 7a,b), which are the sites of most active deep convection in
the winter months (e.g. Lozier, 2010, 2012) and are characterised by ventilation anomalies that persist for longer than in the
Southern Hemisphere (Figure 8). In other words, anomalies near the time of dye injection in the Southern Hemisphere result
in smaller longer-term changes than in the Northern Hemisphere (i.e. the "ventilation persistence" is lower).

There is coherent structure in the residuals of the correlation between Northern Hemisphere ventilation thickness close to
the time of dye injection and 25 years later and the residuals deviate from the trend (rise) for the vintages corresponding to
the early 90's (Figure 10b). We expect that the role of surface buoyancy forcing through air-sea fluxes, as well as modes of
climate variability that may affect convection in the subpolar North Atlantic (e.g. North Atlantic Oscillation - NAO - or Atlantic
Multidecadal Variability - AMV) should already be captured by the correlation itself (Figure 10a). However, these will, in turn,

also affect the background circulation. We examine whether there is a relationship between the residuals and the strength of the large-scale circulation (AMOC). The hypothesis is that a stronger circulation (AMOC) might also be linked with moving

water more effectively away from the ventilation site and reducing the rate at which dye is returned to the mixed layer and thus removed from the system. We consider the correlation between the residuals in the Northern Hemisphere and the AMOC at 26°N and 1000 m depth (to be comparable to estimates for the RAPID array; Figure 13). There is a correlation ($R^2 = 0.45$) between the residuals and the AMOC in the year of injection, while that with the AMOC 25 years later is much weaker ($R^2 = 0.2$).

The mechanism behind the connection that we find between the strength of the AMOC at 26°N and the residuals of the correlation for the Northern Hemisphere (Figure 13) is not fully clear. To check this potential link further, we also correlated the AMOC in the year of injection with the absolute values for ventilation thickness after 25 years, rather than the residuals of the correlation. While both correlations are significant, the correlation of the AMOC with the residuals describes more of the variance (Figure 13) than that with the ventilation thickness itself ($R^2 = 0.37$; not shown). We also tested the correlation of the

residuals with the AMOC at higher latitudes (e.g. 45°N; not shown), but despite being closer to the sites of deep convection in the subpolar North Atlantic, the correlation is much weaker. It is, however, harder to assess and interpret AMOC changes at these higher latitudes, especially in depth rather than density space (Zou et al., 2020; Hirschi et al., 2020). There are also large differences in the simulation of the magnitude and variability of the (A)MOC across different numerical models, both at lower and higher resolutions (e.g. Hirschi et al., 2020; Li et al., 2019); as well as substantial discrepancies in simulating the mixed

layer depth, especially in high latitude regions, and in the representation of water masses, as clearly shown for Labrador Sea Water (Li et al., 2019). Both in models and observations, the relationship between the hydrographic variability in the subpolar North Atlantic and the AMOC has also been shown to quickly deteriorate downstream of the Labrador Sea (e.g. Zou et al., 2019; Li et al., 2019).

There is also some structure in the residuals of the correlation for the Southern Hemisphere (Figure 10d), perhaps reflecting

how ventilation here appears to be stronger before the 80's and consistently lower in the later period (Figure 9a). This could highlight background changes in the strength of the subtropical gyres, driven by changes in the Southern Hemisphere winds, affecting SAMW formation (e.g. Jones et al., 2016; Waugh et al., 2019; Meijers et al., 2019). While there are more opportunities to test the model's performance in the Northern Hemisphere, given the richness of observations from the North Atlantic subpolar gyre, recording years of weaker and strong convection, a more prominent focus on Southern Hemisphere processes

and changes in SAMW would be the desirable outcome of a future study exploiting our setup. This could be achieved for the more recent vintages (from the 90's onwards; only partly used here, due to the focus on the 3 to 25 years time scales and the constraints due to the length of the forced simulation). In fact, recent hydrographic observations have highlighted ventilation changes due to wind forcing and atmospheric modes (Talley et al., 2016; Meijers et al., 2019) and other modelling studies have explored the different pathways followed by SAMWs and their sensitivity to wind changes (Jones et al., 2016; Waugh et al.,

2019) providing better estimates of their combined effect on heat and carbon uptake in the Southern Hemisphere, which could be integrated with our dye tracers.

Finally, while it would also be desirable to perform longer simulations with passive tracers and assess their uptake on centennial time scales, our results highlight how this would be problematic. In fact, the tracer's pathways, inventory and distribution will be strongly dependent on the initial surface forcing, when the tracer enters the ocean, especially for the Northern Hemisphere (see Figures 11 and 12). This means that the cumulation of the inventory of a simulated tracer over time will provide an aliased view of its evolution. At the same time, it is at present too computationally-expensive to introduce interannually-varying tracers to resolve the full variability on such long timescales.

## 5 Summary and conclusions

We have used a set of interannually-varying passive dye tracers in an ocean-ice model to explore pathways and time scales of ocean ventilation. This is a computationally-expensive approach, but it allows us to fully capture the pathways of subducted waters after they leave the mixed layer.

The Southern Hemisphere shows more variability in ventilation thickness just after the tracer's injection, while the Northern Hemisphere is characterised by higher variability in the 25-year inventory, highlighting different ventilation "efficiencies" and timescales. Subducted waters are exported faster in the Northern than in the Southern Hemisphere, but the correlation between ventilation thickness after three and 25 years is strong in both hemispheres. This means that the strong interannual variability in the initial surface forcing will dominate on longer time scales, and largely override the different processes that drive the uptake and export of passive tracers.

Our results highlight the key role played by surface forcing near the time when a tracer enters the ocean in setting the long-term variability of its inventory and determining the pathways and timescales of their uptake by the ocean. This has important implications for the interpretation of observations that only capture snapshots of the circulation and also offers potential to forecast changes in the pathways and uptake of tracers by the ocean, such as anthropogenic carbon and heat (e.g. Zanna et al., 2019; Bronselaer and Zanna, 2020), with important consequences in estimating changes in their inventories and long-term storage and sequestration.

Given the deficiencies of our coarse-resolution model, it would be desirable but highly computationally expensive to apply our method to a fully eddy-resolving setup. The use of higher resolution models would prove challenging even with a smaller subset of interannually varying tracers. Comparisons between different models would also be insightful, but once again not easy to achieve due to the computational costs; in this case, the use of offline Lagrangian trajectories could be a satisfactory compromise. Finally, it would be feasible, but once again computationally very expensive, to apply our methodology at 1° resolution with tracers that are released from a set of surface patches that represent the source regions of different water masses (following the methodology of e.g. Khatiwala et al., 2009, 2012, 2013) that has been used for the interpretation of derived quantities such as anthropogenic carbon.

*Author contributions.* AM, GN and EM designed the ocean-ice simulations with tracers. AM ran the simulations and analysed the data. LC carried out the analysis for the data shown in Figure 1. All authors contributed to the interpretation of the data and to the writing of the manuscript.

*Competing interests.* The authors declare no competing interests

*Acknowledgements.* AM, GN, LC, EM were supported by Natural Environment Research Council grant NE/P019293/1 (TICTOC) and EM was also supported by European Union Horizon 2020 grant 817578 (TRIATLAS).

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

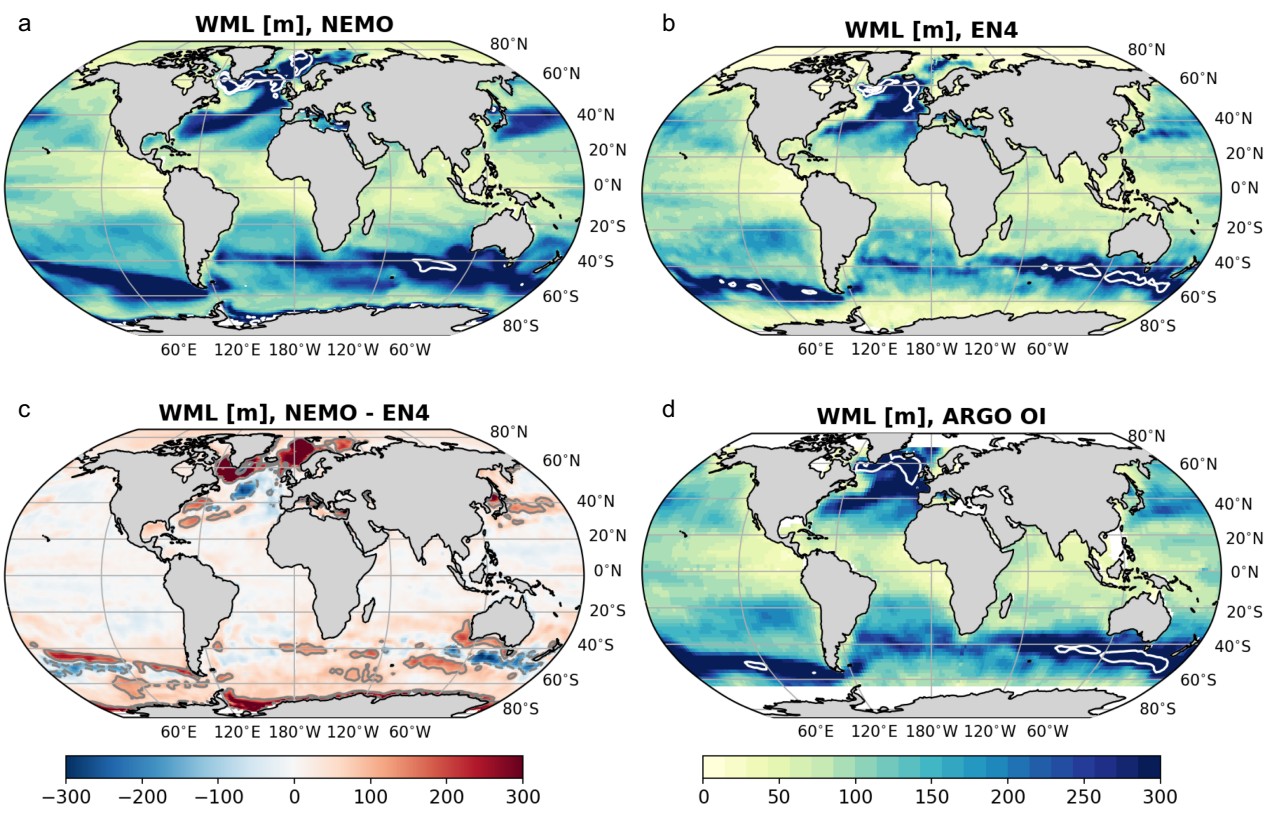

**Figure 1.** Winter (i.e. temporal maximum) mixed layer depth (m) between 2004 and 2017 in NEMO (a), EN4 (b) and Argo (d) and the difference between NEMO and EN4 (c). Note that the temporal resolution is monthly for NEMO and EN4, while it is every 10 days for Argo.

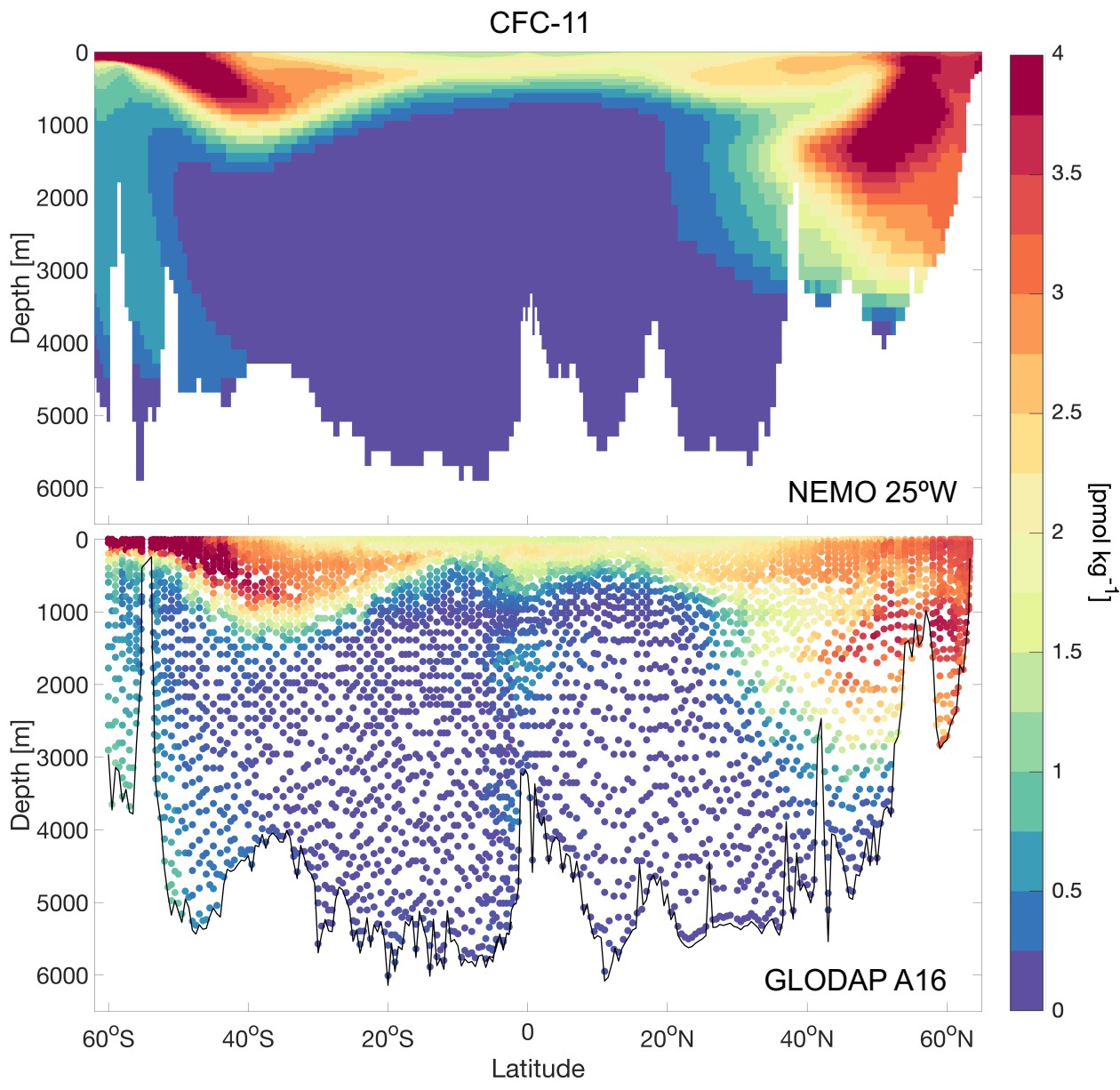

**Figure 2.** CFC-11 distribution in the model along a mid-Atlantic section at 25°W (a) and observations from the GLODAP dataset along the WOCE/GO-SHIP A16 line.

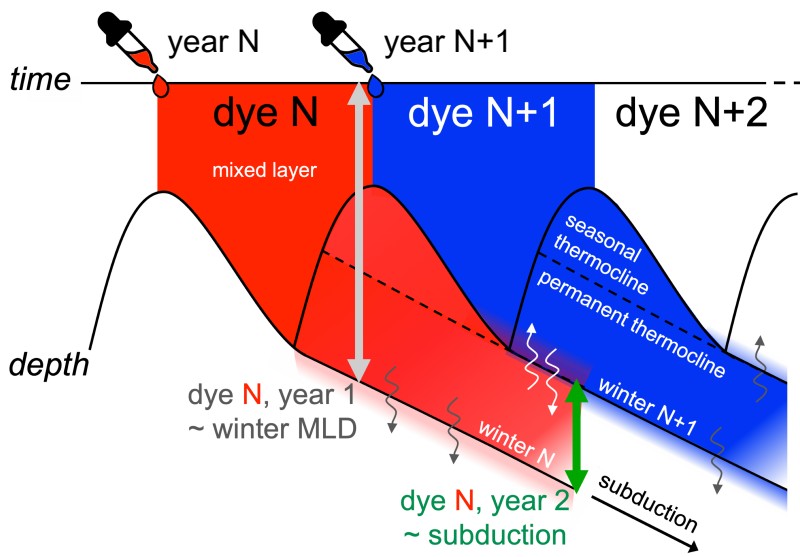

**Figure 3.** Schematic representation of the injection of dye tracers in the model.

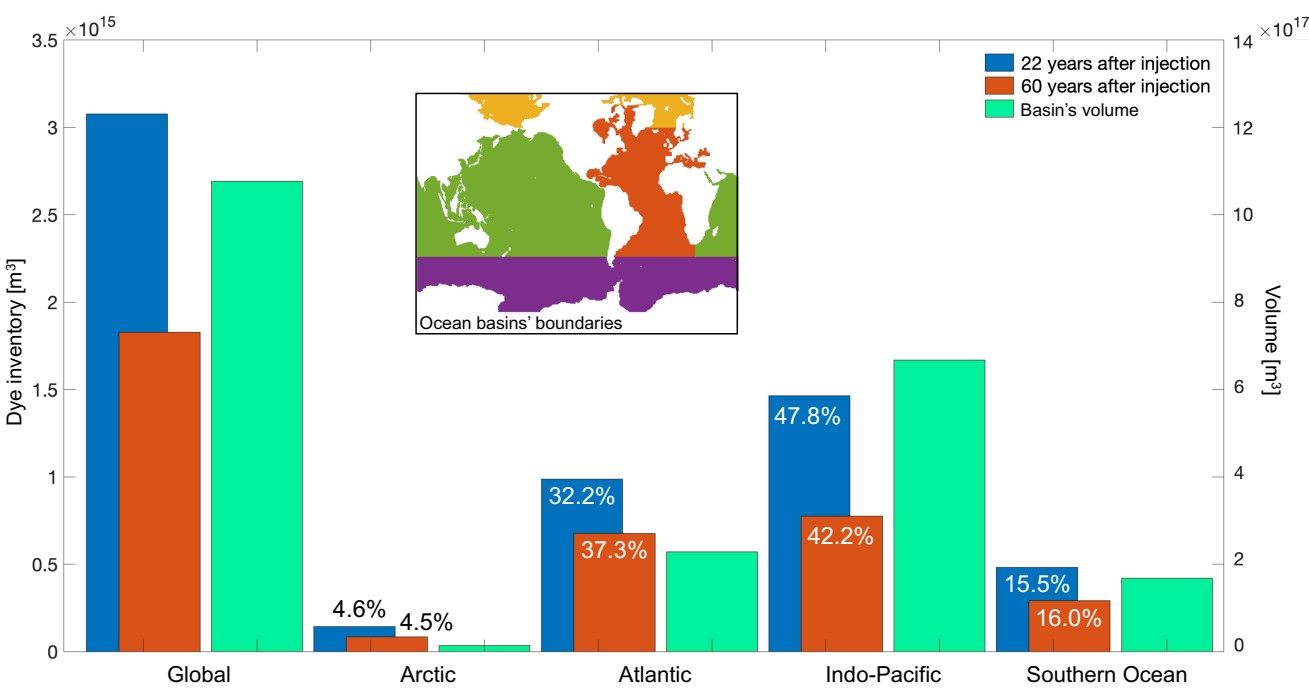

**Figure 4.** Dye inventory for each ocean basin and globally 22 years after injection (in 1980 for the 1958 vintage; blue bars, left y-axis) and after 60 years (in 2017; orange bars) and the volume of each basin (green bars, right y-axis). Numbers indicate the respective percentage of the global inventory for each ocean basin and the inset shows how the boundaries of the basins have been defined here.

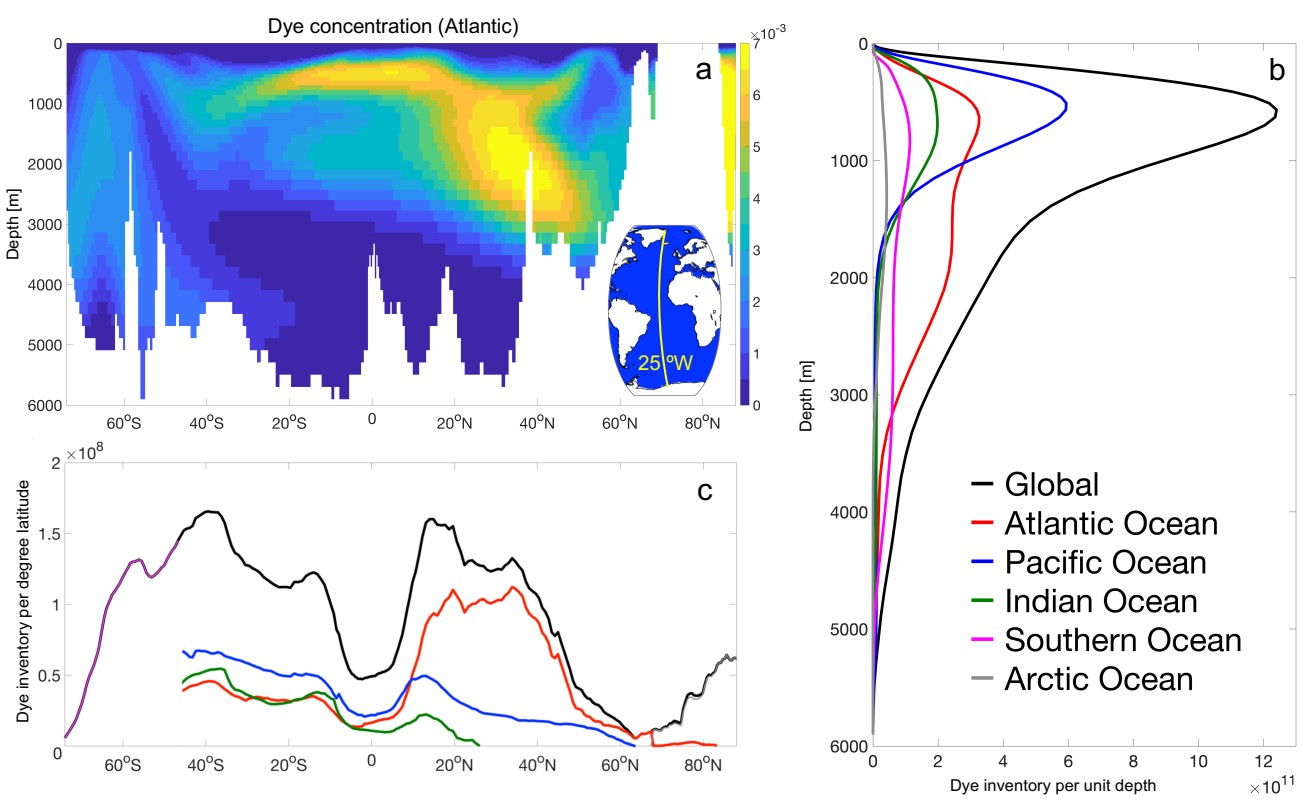

**Figure 5.** Mid-Atlantic cross section (25°W, as shown in the inset) of dye concentration (a) and its distribution as a function of depth (b) and latitude (c) globally and for all ocean basins at the end of the simulation (2017), after 60 years of dye injection (1958 vintage).

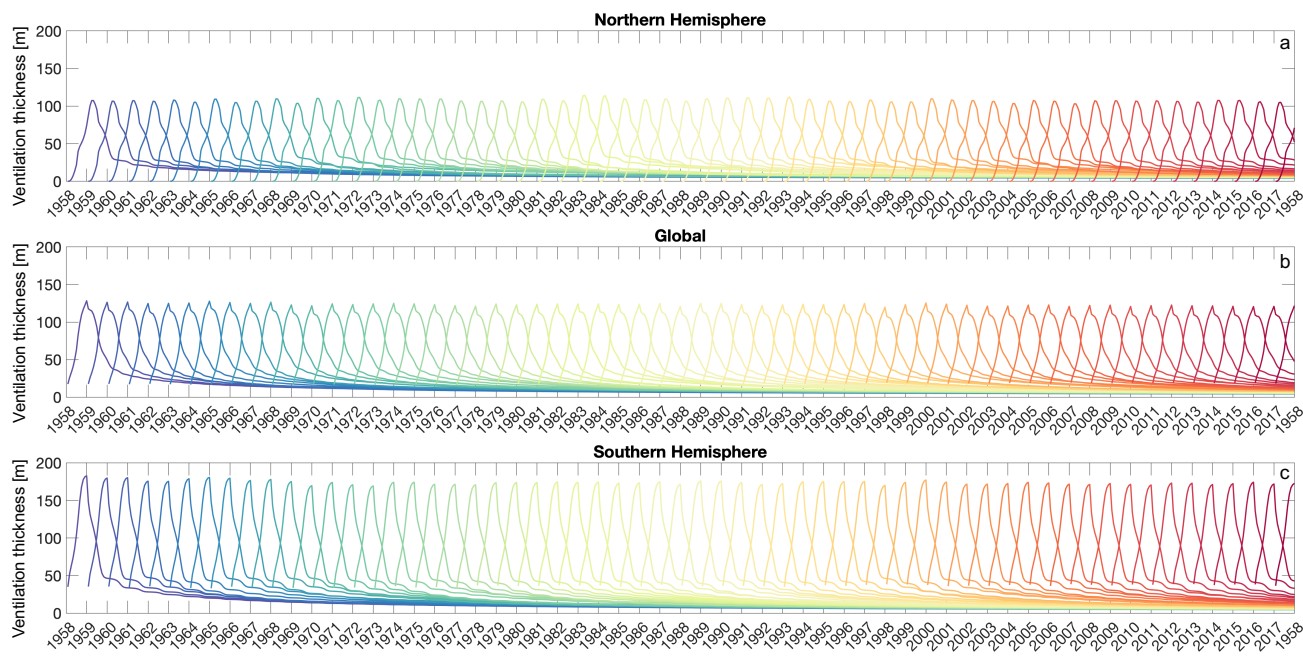

**Figure 6.** Ventilation thickness for all dyes/vintages (1958-2017) in the Northern Hemisphere (a), as a global average (b) and in the Southern Hemisphere (c).

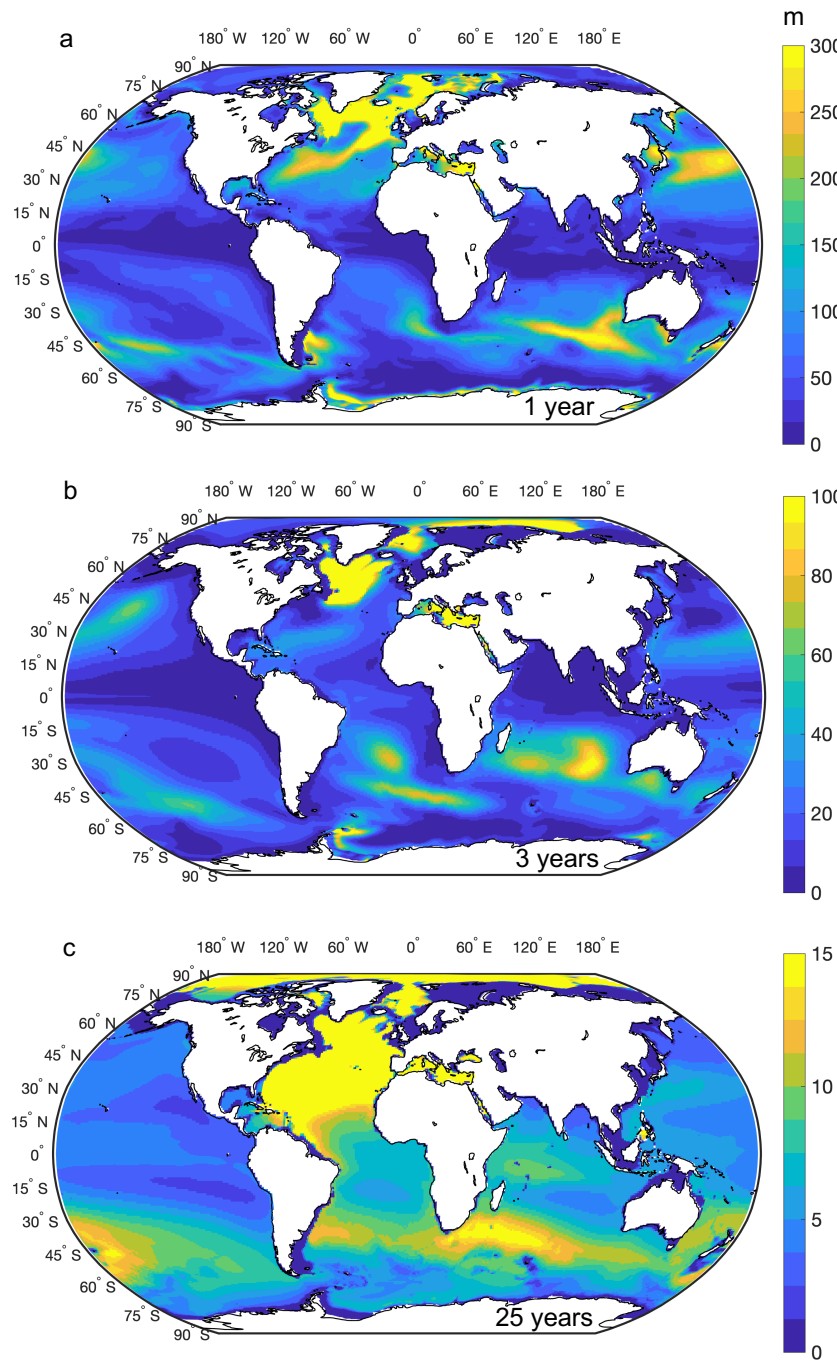

**Figure 7.** Ventilation thickness 1 year (a), 3 years (b) and 25 years (c) after the dye injection for one of the vintages.

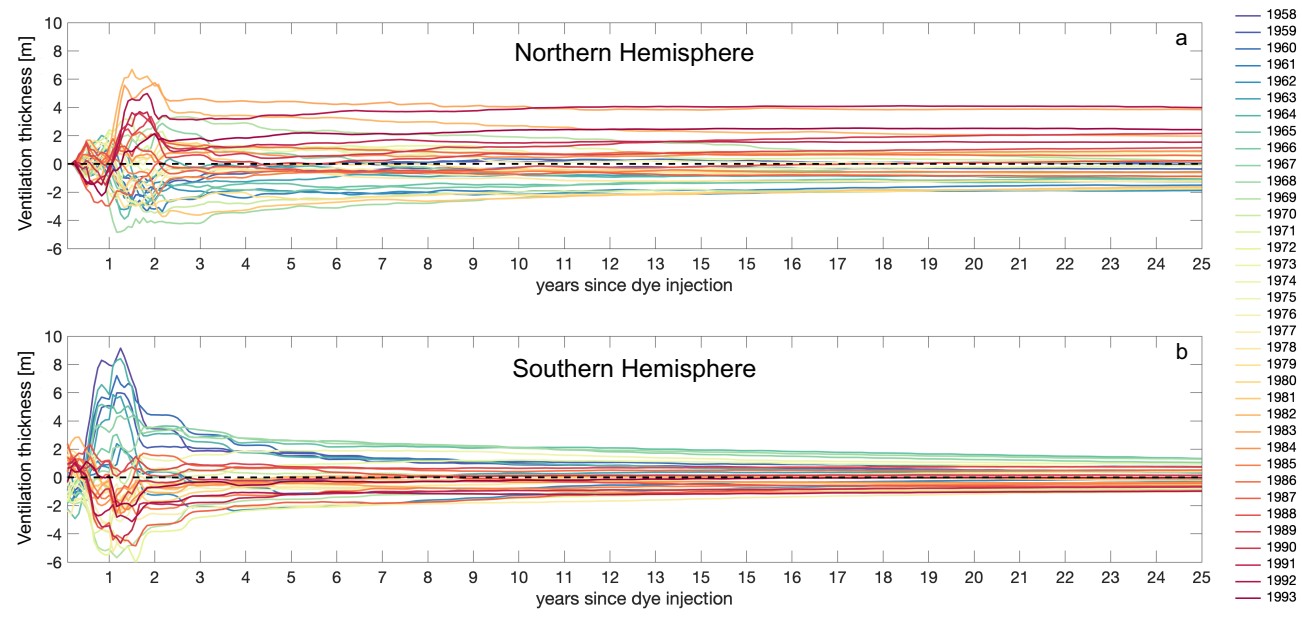

**Figure 8.** Ventilation thickness anomaly (from the 1958-1993 mean) for 36 of the 60 dyes (1958 to 1993 vintages) from injection to 25 years later.

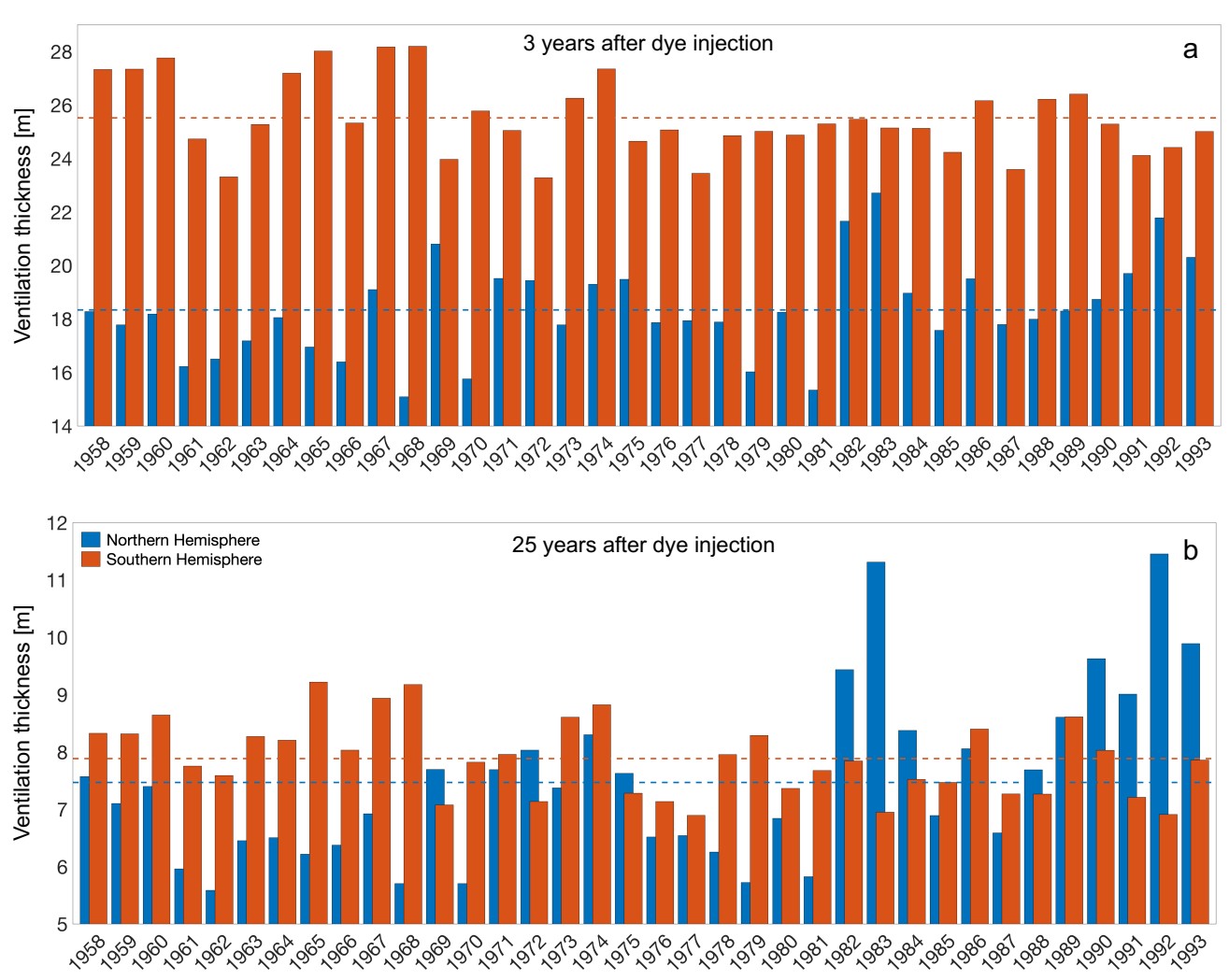

**Figure 9.** Ventilation thickness for both hemispheres and for the same 36 dyes of Figure 8, 3 years (a) and 25 years after injection (b). Note that the scale in different in the two panels and it does not start from zero.

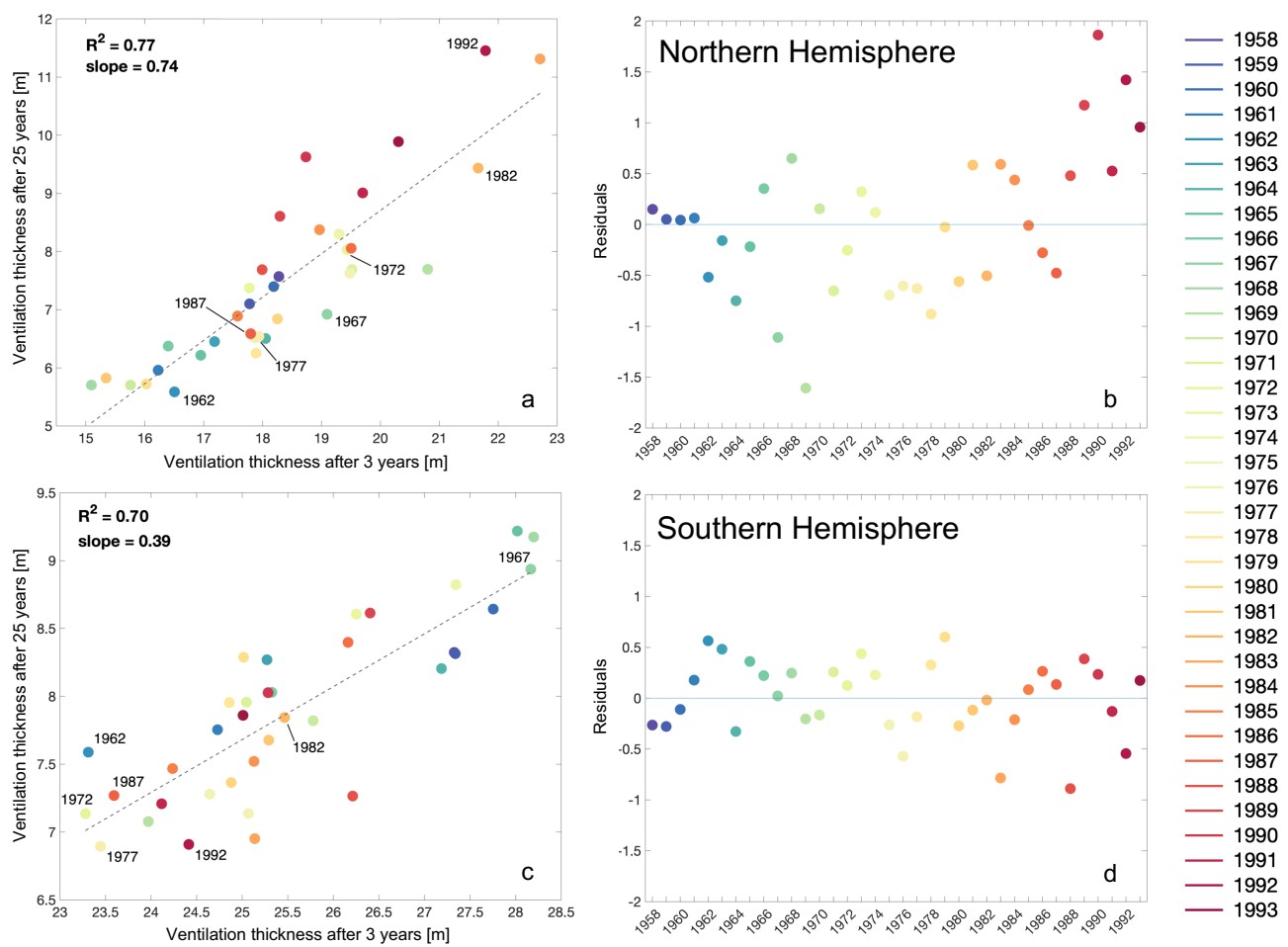

**Figure 10.** Correlation between ventilation thickness after 3 years and after 25 years in the Northern (a) and Southern Hemisphere (c) and the residuals of the correlation in both hemispheres (b, d, respectively). The years of injection for the vintages shown in Figure 9 are highlighted for both hemispheres (a, c).

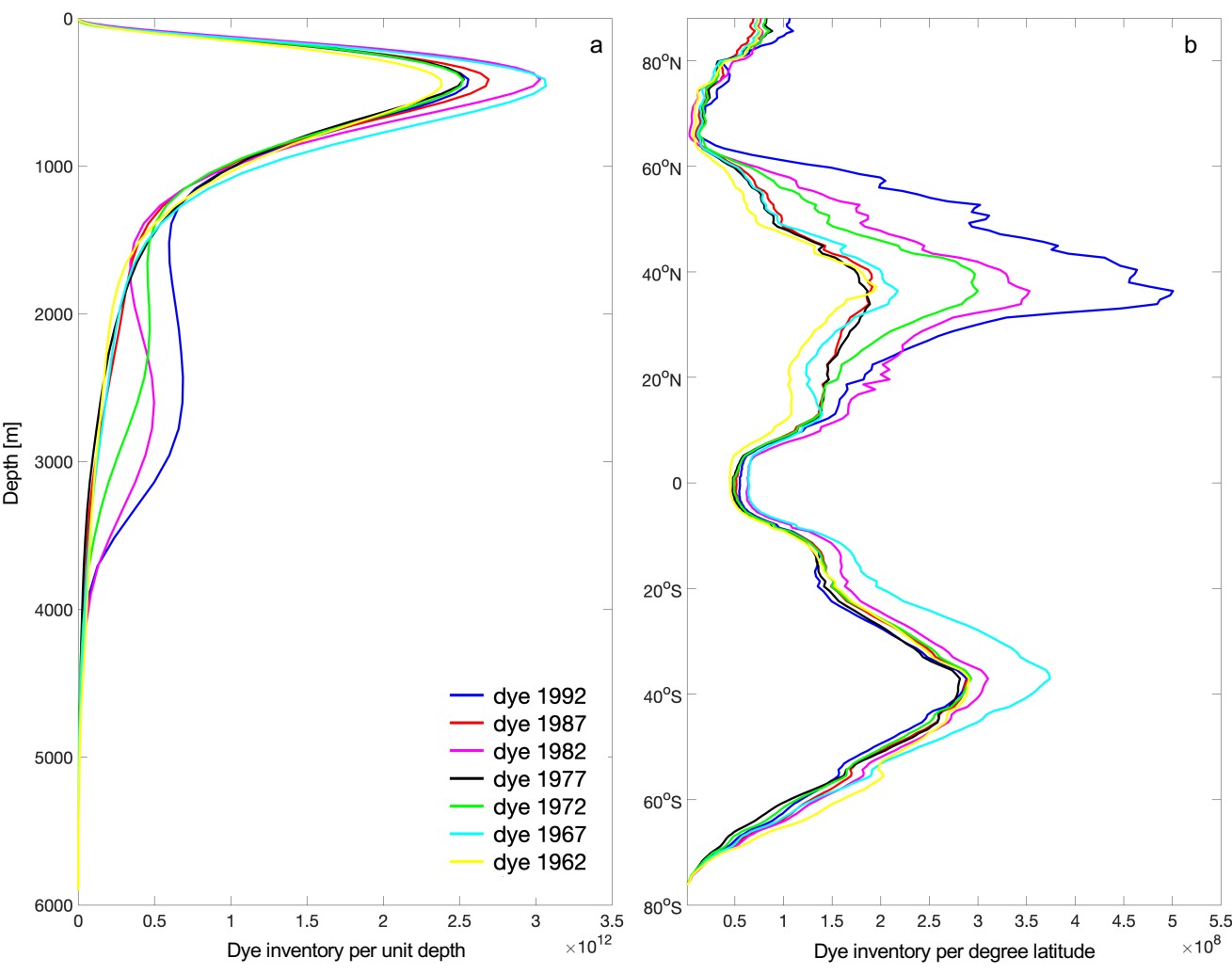

**Figure 11.** Global tracer concentration 25 years after injection for seven different vintages as a function of depth (a) and latitude (b).

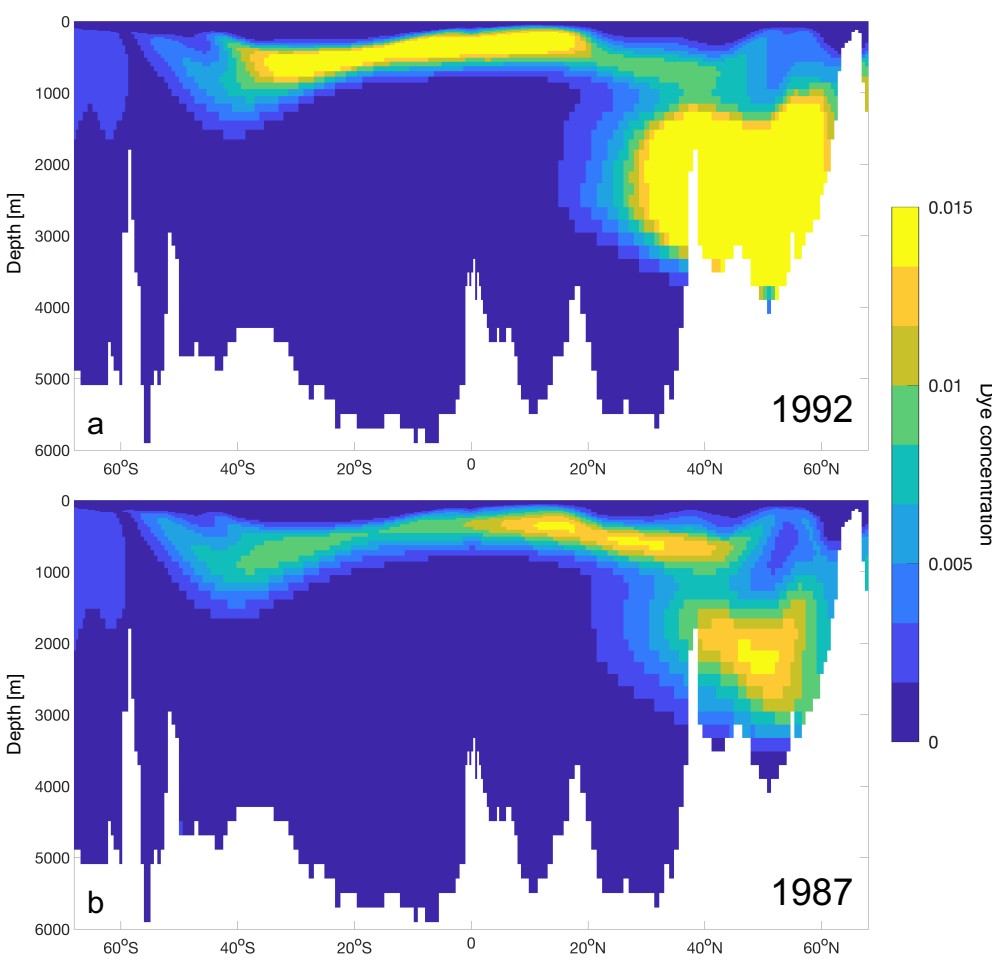

**Figure 12.** Tracer concentration 25 years after injection for two of the vintages, one that was injected in a year of strong Northern Hemisphere convection (a) and a weak one (b) along a mid-Atlantic north-south section (as shown in the inset of Figure 5a).

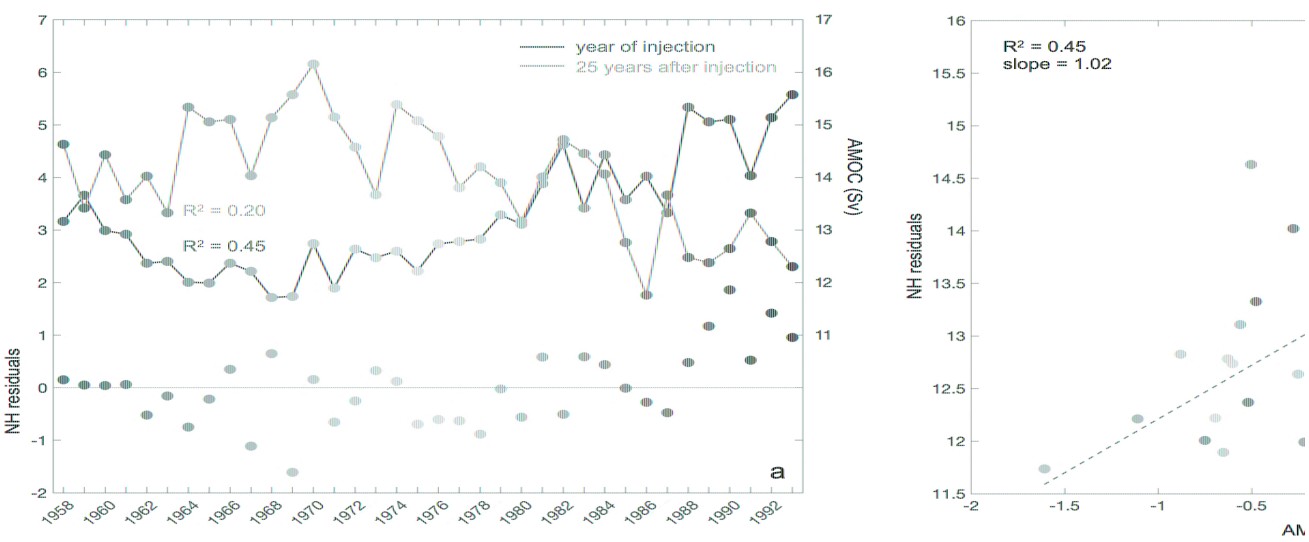

**Figure 13.** Timeseries of AMOC at 26°N at 1000m and Northern Hemisphere residuals of the correlation in Figure 10b (a) and correlation of the AMOC against the residuals (b) from panel a.