# Peer review of "Surface atmospheric forcing as the driver of long-term pathways and timescales of ocean ventilation"

_Ocean Science, 2021_

## Author Response (AR1)

**Reviewer 1**

We thank the reviewer for the very positive comments and for the helpful suggestions, which we have now addressed in our revised manuscript.

**Major Comments**

1) I have a couple of concerns regarding the "ventilation depth" metric. First I am not sure "ventilation depth" is the best terminology as it gives the impression that this related to a measure of how deep the ventilation is occurring in the water column. At least as I was reading the text that is what I was tempted to think and I had to remind myself it was not a physical depth.

We have now clarified in the text that this is not a physical depth (page 7, lines 198-202) and renamed this metric as "ventilation thickness" throughout the manuscript.

Second, and partly related, I am not sure it is best to start with this metric. I think it would be better to start with metrics / plots like figure 9. These plots give a much better impression of how the tracer had infiltrated the ocean interior / where the tracer is. These could then be related to the tracer volume per area metric, which could then used as the summary metric.

We have re-ordered some of the existing figures and introduce earlier in the manuscript metrics and plots that show the distribution of the tracer more explicitly – e.g. see the new Figure 5 and the modified Figure 4. Ventilation thickness is now introduced only from Figure 6 onwards.

2) Most of the focus is on NH but there have been well documented changes in SH winds (eg SAM trends). I think it would be good include some discussion on whether there is any signal of these trends in your ventilation tracers.

Related to this, several recent papers (Jones et al. 2016, 2019, Waugh et al. 2019) indicate that the rate that young waters is transported into the southern permanent pycnocline depends not only on the rate at which they are subducted but also on the speed at which the gyres circulate. Is this inconsistent with your result that the ventilation near the time of dye injection sets the long-term variability for the dye inventory?

The focus on the NH is due to the known strong link between surface atmospheric forcing and deep-water formation in the subpolar North Atlantic and the availability of observations that can be compared more directly to our results. Such as those pinpointing years of particularly strong convection in the Labrador and Irminger seas. But we are now mentioning Southern Hemisphere processes more explicitly when discussing longer-term/larger-scale signals and trends, rather than only focusing on the potential impact of the AMOC, and we are now referencing a few relevant studies, including those suggested by the reviewer (page 12, lines 379-391).

We are indeed working towards a separate study addressing changes related to SAMW and will make more extensive use of the vintages from the 90's and 2000's, when more observations are also available for these regions.

**Minor Comments**

Line 120 Question marks in Sv definition.

Line 350 "fficienciesnd"

We have corrected these.

**Reviewer 2**

We thank the reviewer for the very positive comments and for the helpful suggestions, which we have now addressed in our revised manuscript.

**General**

Model vs reality: my biggest criticism is that I found it frustrating that we aren't really given the information to know how to relate the results of the study to the real ocean. The study is of a model, but one forced with re-analyses for a specific modern time interval, suggesting we ought to be able to relate the results to the present day ocean. In much of the paper the authors are discussing inter-annual and decadal variability, which they would like to relate to the real world, but it's hard to tell how much faith to put in this. There are some important model aspects that don't accord with the real ocean -- we are told for example that it captures N. Atlantic and Southern mode water formation but that there is too-deep ventilation in the N. Atlantic. In the Southern hemisphere, while there is SAMW formation, there appears to be little or no AABW formation. So to what extent should we view it as just a model study, and to what extent a simulation of ventilation in the real ocean?

What would really clarify this for me would be a comparison with how this model's simulation of say, CFC distribution compares with that in the real ocean. I think it ought to be possible to do this without doing any more model runs since, as the authors point out, their tracers constitute a set of numerical Greens functions for transfer from the surface into the interior for each year of this 60 year period. Even if this is not possible, I think there should be a section in the paper dealing more explicitly with the issue of the extent to which the model results are interpretable in the real ocean.

Our model runs actually included CFCs, input in the last 60-year forcing cycle. We have now included a comparison between modelled CFC-11 and observations from the GLODAP dataset for the mid-Atlantic section used in the study (Figure 2), which shows a good degree of agreement (page 5, lines 146-160. We also now explain specifically how it is the sum of dye tracers rather than the dye tracers themselves that should be related directly to CFC distributions (page 6, lines 191-194).

The model also shows a good response to the forcing in its representation of the variability of strong/weak convection in the North Atlantic subpolar gyre, as discussed in section 3.2 (see Figure 9).

As discussed in section 4, our results correlating ventilation thickness on different timescales and highlighting the dominant role of surface forcing in setting the evolution of the tracers' distribution and inventory, are not affected by the model biases. However, we describe the shortcomings of coarse resolution simulations and how/where these may affect tracers' distributions. The comparison with observations (MLD, CFCs, interannual variability of convection in the Labrador Sea) provides additional confidence in how this model is able to represent the processes that are relevant to our study and to capture their interannual variability.

**Minor:**

Use of "Ventilation depth": The natural interpretation of this term would be the depth to which the tracer has penetrated. However, in the paper it is used to mean something rather different and quite specific: the definition makes clear (Line 176) that it is the column integral of the tracer, not an actual depth. However the tracer is defined as dimensionless so that the column integral has dimensions of depth. I would suggest at least a sentence after the definition to emphasise what is, and is not, meant by the term. Maybe the terminology should be changed – why not call it the "column integral" for instance?

We have clarified the definition and specified that this metric represents the column integral (lines page 7, lines 198-202). As proposed in addressing a similar comment by Reviewer 1, we have now renamed this metric "ventilation thickness" throughout the manuscript.

Line 251: "Ventilation appears to be twice as effective in the Northern than the Southern hemisphere". This is a fascinating finding that deserves a good deal more investigation and some explanation. Is it due to the shortcomings of the model, or is it actually true in the real world? Even if it's a model effect it is worth understanding better. The comparison with a real tracer such as CFCs that I suggested above would help here.

We have rephrased this as "ventilation persistence", to highlight how subducted waters are exported (and isolated) away from deep mixed layer regions faster in the Northern than the Southern Hemisphere, meaning that anomalies near the time of dye injection in the Southern Hemisphere result in smaller longer-term changes than in the Northern Hemisphere (page 11, lines 345-352).

Discussion line 317 and following on strength of AMOC and ventilation: I'm rather unclear as to why the authors concentrate on the residuals of fig 7, e.g we are not shown the correlation with the actual ventilation at 25 years and the AMOC, but rather the degree to which that is different (mostly, higher) than expected from the correlation with the ventilation at 1 year. I find it hard to interpret that finding (and clearly the authors do too: they say (line 330) "the mechanism is not fully clear" – I would say that is an understatement). Is there no correlation between the actual amount of ventilation at 25 years and the AMOC at the time of ventilation? (by eye I would think there probably is?)

We have performed the suggested correlation and found it to be weaker than that with the residuals (R2: 0.37 and 0.45, respectively) but both correlations are significant. This is now mentioned in the text (page 12, lines 365-369). We do not feel we can provide additional insight on this mechanism at this stage, but it will be worth re-examining in a future study that we are working on and that focuses on the North Atlantic subpolar gyre.

**Detailed comments:**

Line 107: The FCT numerical scheme used has "moderate numerical diffusion". The authors don't give an estimate of "moderate" is, but for detailed tracer studies, numerical mixing should be small. It would be good to give a rough estimate, and they might comment on how this would affect their results.

We have updated the description of the numerical scheme (page 4, lines 104-113). Numerical diffusivity is not easy to diagnose, particularly for a flux-corrected scheme whose diffusivity depends on the tracer distribution. Recent work (Megann, 2018) has suggested that this FCT scheme can give equivalent numerical vertical diffusivity of  $1-3 \times 10^{-4} \text{ m}^2/\text{s}$  in global simulations run at  $1/4^\circ$  horizontal resolution, associated with strong oscillatory vertical velocities up to ~5 x  $10^{-5}$  m/s. Such velocities, 50 times larger than typical Ekman pumping values, are however not seen at our 1° resolution, so numerical diffusivity should be lower than these values, given that diffusivity scales at worst as  $\Delta z.w$  and our vertical resolution is the same as in the  $1/4^\circ$  model diagnosed by Megann (2018).

Reference: Megann, A., 2018. Estimating the numerical diapycnal mixing in an eddy-permitting ocean model. *Ocean Modelling*, *121*, pp.19-33.

**Line 225: some typos here.**

We have corrected these.